

# Probabilistic approaches for investigating species co-occurrence from presence-absence maps

Ya-Mei Chang[1], Suman Rakshit[2,3], Chun-Hung Huang[1] and Wen-Hsuan Wu[1]

[1] Department of Statistics, Tamkang University, New Taipei City, Taiwan
[2] School of Electrical Engineering, Computing and Mathematical Sciences, Curtin University, Perth, Western Australia, Australia
[3] Curtin Biometry and Agriculture Data Analytics, School of Molecular and Life Sciences, Curtin University, Perth, Western Australia, Australia

## ABSTRACT

**Background**. In this research, we propose probabilistic approaches to identify pairwise patterns of species co-occurrence by using presence-absence maps only. In particular, the two-by-two contingency table constructed from a presence-absence map of two species would be sufficient to compute the test statistics and perform the statistical tests proposed in this article. Some previous studies have investigated species co-occurrence through incidence data of different survey sites. We focus on using presence-absence maps for a specific study plot instead. The proposed methods are assessed by a thorough simulation study.

**Methods**. A Chi-squared test is used to determine whether the distributions of two species are independent. If the null hypothesis of independence is rejected, the Chi-squared method can not distinguish positive or negative association between two species. We propose six different approaches based on either the binomial or Poisson distribution to obtain p-values for testing the positive (or negative) association between two species. When we test to investigate a positive (or negative) association, if the $p$-value is below the predetermined level of significance, then we have enough evidence to support that the two species are positively (or negatively) associated.

**Results**. A simulation study is conducted to demonstrate the type-I errors and the testing powers of our approaches. The probabilistic approach proposed by Veech (2013) is served as a benchmark for comparison. The results show that the type-I error of the Chi-squared test is close to the significance level when the presence rate is between 40% and 80%. For extremely low or high presence rate data, one of our approaches outperforms Veech (2013)'s in terms of the testing power and type-I error rate. The proposed methods are applied to a tree data of Barro Colorado Island in Panama and a tree data of Lansing Woods in USA. Both positive and negative associations are found among some species in these two real data.

Corresponding author
Ya-Mei Chang, yamei628@gmail.com

## INTRODUCTION

Incidence-based data are common in biology and ecology (*Gotelli & Chao, 2013*). This kind of data ease a lot of fieldwork and reduce cost because only presence or absence needs
to be recorded. It may be the only option in some circumstances, especially when it is hard to count individual members of a species. The main objective of this article is to examine species co-occurrence (also called "co-existence") by using presence-absence maps only. In particular, the presence-absence map of two species can be used to construct the two-way contingency table containing the four frequencies related to the presence and absence combinations of two species, and these frequencies would be sufficient to compute the six new test statistics proposed in this article.

The investigation of species co-occurrence plays an important role in studying gene expression, community assembly, local–regional species diversity relationship, and meta-community concept (*Gotelli & Ulrich, 2010*; *Veech, 2014*). The co-occurrence patterns are typically categorised into three associations: (i) positive, (ii) negative, and (iii) random. Methods of analysis for these data can broadly be divided into the classes of matrix-level and pairwise approaches. The matrix-level approaches investigate the entire assemblage as a unit of analysis, whereas the pairwise approaches focus on determining patterns of positive, negative, or random association between two species. The methods proposed in this article belong to the pairwise class of approaches.

Several pairwise approaches have been proposed in the last twenty-five years (*Veech, 2006*; *Veech, 2013*; *Sanderson, 2000*; *Sanderson, Diamond & Pimm, 2009*; *Sfenthourakis, Giokas & Tzanatos, 2004*; *Sfenthourakis, Tzanatos & Giokas, 2006*; *Gotelli & Ulrich, 2010*; *Pitta, Giokas & Sfenthourakis, 2012*). See *Veech (2014)* for an overview of pairwise approaches used in various applications of species co-occurrence. When determining species co-occurrence, it is crucial to avoid misinterpreting it as species interaction or causality. The analysis results of *Cazelles et al. (2016)* revealed the difficulty of the interpretation of species interactions from co-occurrence data. *Blanchet, Cazelles & Gravel (2020)* mentioned that both theory and experimental evidence support the idea that ecological interactions may affect co-occurrence, but co-occurrence is not evidence of ecological interactions. *Dormann et al. (2018)* provided ten questions to guide interpretation and avoid false conclusions.

In this article, we use the Chi-squared statistic to test the null hypothesis that the distributions of two species are independent; see *Lindgren* (*2017*, Ch 10) for an overview of the Chi-squared test of independence. If the null hypothesis is rejected, the Chi-squared method can not determine whether there is a positive or negative association. We propose a few different combinations of random variables under binomial and Poisson distributions to construct several test statistics; consequently, six different approaches are developed to compute $p$-values for inferring positive (or negative) association. If the computed $p$-value is smaller than the chosen level of significance (typically chosen to be 5%), we conclude that there is sufficient evidence to reject the null hypothesis and support the alternative hypothesis of positive (or negative) association between two species.

To assess the performance of the six newly proposed tests, we perform an extensive simulation study (details are given in the next section). Moreover, we have included the probabilistic approach proposed by *Veech (2013)* in the simulation study as the benchmark. The results of the simulation study show that one of our approaches outperforms *Veech*

*(2013)*'s approach in terms of statistical power. *Veech (2013)* investigated species co-occurrence through incidence data of different survey areas. We focus on using presence-absence maps on a certain study plot instead. To illustrate the applicability of our proposed methods, we have analysed two real-life datasets of the presence-absence of several tree species from Barro Colorado Island in Panama and Lansing Woods in the USA.

The rest of this article is organized as follows. The following section introduces the six different approaches proposed in this article and the probabilistic approach proposed by *Veech (2013)*. The simulation settings and the two example datasets of tree species are also presented in the next section. The results of the simulation study are reported in section Results. The Results section also illustrates the analysis of the two example datasets. The article ends with a discussion in the final section.

## MATERIALS AND METHODS

### Example datasets

We have analysed two real-life datasets using several pairwise approaches to determine the species-by-species co-occurrence patterns of tree species in Panama and the USA. The first dataset consists of the tree locations, recorded in 2015, from Barro Colorado Island (BCI) in Panama; it is downloaded from "https://forestgeo.si.edu". The dataset contains 248,835 tree locations capturing 296 tree species of Panama inside a study area of 50 hectare ($500 \times 1000 \ m^2$). In this article, we have selected eight most abundant species to investigate their co-occurrences. See *Condit (1998)*, *Hubbell et al. (1999)*, *Condit et al. (2019b)*, and *Condit et al. (2019a)* for further details about this dataset.

The second dataset is the famous *Lansing* dataset from the R package **spatstat.data**. This dataset provides the locations of 2,251 trees and their botanical classification into six types: black oak, hickory, maples, red oak, white oak, and miscellaneous. These data were collected by *Gerrard (1969)* from a plot in Lansing Woods, Clinton County, Michigan, USA, that measures 924 ft × 924 ft (19.6 acre). The original plot size has been rescaled to the unit square.

### Statistical tests

Let $W \subset \mathbf{R}^2$ denote the study area. We divide the study area into a regular grid, and let $N$ denote the total number of grid cells. We are interested in investigating the association between species A and species B. Let $N_a$ be the number of cells in which species A is present and $N_b$ be the number of cells in which species B is present. Under the randomness assumption, the expected number of cells in which both species A and B are present is $E_1 = N_a N_b / N$, in which neither of them is present is $E_2 = (N - N_a)(N - N_b)/N$, in which species A is present but species B is absent is $E_3 = N_a(N - N_b)/N$, and in which species A is absent but species B is present is $E_4 = (N - N_a)N_b/N$. For each $E_i$, $i = 1, \ldots, 4$, let $O_i$ denote the corresponding random variable. Therefore, the random variables $O_1$, $O_2$, $O_3$, and $O_4$ represent the number of grid cells (out of $N$ cells) containing both species $A$ and $B$, containing neither of them, containing only species $A$, and containing only species $B$, respectively. Furthermore, let $o_i$, $i = 1, \ldots, 4$, denote the observed value corresponding to the random variable $O_i$. A Chi-squared test is used to test the null hypothesis that the

occurrences of species A and B are independent, and the corresponding test statistic can be computed as

$$\chi^2 = \sum_{i=1}^{4} \frac{(o_i - E_i)^2}{E_i}. \tag{1}$$

Under the null hypothesis of independence, the test statistic Eq. (1) follows the Chi-squared distribution with 1 degree of freedom (d.f.). If at least one expected frequency $E_i$ is smaller than ten, then the above test statistic is replaced by the Yates' corrected version (*Yates, 1934*):

$$\chi^2_{\text{Yates}} = \sum_{i=1}^{4} \frac{(|o_i - E_i| - 0.5)^2}{E_i}. \tag{2}$$

When $(|o_i - E_i| - 0.5)$ is negative in Eq. (2), it is set equal to zero. The main disadvantage of the Chi-squared test approach is the lack of information. When the null hypothesis is rejected, we need some other method to investigate whether the alternative hypothesis of positive (or negative) association between the two species can be supported.

To develop new approaches for testing associations between species, we utilise the following key results. Under the assumption that every individual is randomly distributed in each cell with equal probability, the random variable $O_i$ follows the binomial distribution with $N$ "trials" and "success" probability $E_i/N$, which is expressed as $O_i \sim \text{Bin}(N, E_i/N)$. If $N$ is large and the "success" probability $E_i/N$ is small, then $O_i$ is approximately distributed as Poisson distribution with expected value $E_i$, and this is expressed as $O_i \sim \text{Poi}(E_i)$ for $i = 1, \ldots, 4$. Similarly, if the randomness assumption holds true, for large $N$ and small $E_i$ ($i = 1, \ldots, 4$), we also have $O_i + O_j \sim \text{Poi}(E_i + E_j)$ for $i \neq j$.

We use the random variables $O_i$, $i = 1, \ldots, 4$, to construct six new test statistics for testing non-random associations between two species. If the occurrences of species A and species B are positively associated, $O_1$ and $O_2$ tend to be large, but $O_3$ and $O_4$ tend to be small. In contrast, if they are negatively associated, $O_1$ and $O_2$ tend to be small, but $O_3$ and $O_4$ tend to be large. For testing positive association, we use the following five rejection rules: (i) $O_1$ is large, (ii) both $O_1$ and $O_2$ are large (iii) $O_1 + O_2$ is large, (iv) both $O_3$ and $O_4$ are small, and (v) $O_3 + O_4$ is small. Note that the test statistics for (i), (iii), and (v) are scaler-valued, whereas the test statistics for (ii) and (iv) are random vectors, given by $(O_1, O_2)^\top$ and $(O_3, O_4)^\top$, respectively. We define the test statistics for assessing negative association by replacing the term large with the term small and vice versa in (i)–(v). Because the random variables $O_1, \ldots, O_4$ follow either binomial or Poisson distributions under the null hypothesis, we show below that it is straightforward to compute the $p$-values using our proposed test statistics.

For testing the alternative hypothesis of positive association, we propose the following six methods to compute the $p$-values. Let $C_r^N$ (read $N$ choose $r$) denote the number of combinations of $N$ things taken $r$ at a time. The quantity $C_r^N$ is computed as $N!/(r!(N-r)!)$, where $N!$ is $1 \times 2 \times \cdots \times N$.

- Under binomial distribution, the test statistic $O_1$ is used to compute the $p$-value as follows: $P_1 = \Pr(O_1 \geq o_1 | H_0) = \sum_{j \geq o_1} C_j^N (\frac{E_1}{N})^j (1 - \frac{E_1}{N})^{N-j}$.

- Under binomial distribution, the test statistic $(O_1, O_2)^\top$ is used to compute the $p$-value as follows: $P_2 = \Pr(O_1 \geq o_1 \text{ and } O_2 \geq o_2 | H_0) \approx \prod_{i=1}^{2} \Pr(O_i \geq o_i | H_0) = \prod_{i=1}^{2} [\sum_{j \geq o_i} C_j^N (\frac{E_i}{N})^j (1 - \frac{E_i}{N})^{N-j}]$.
- Under binomial distribution, the test statistic $(O_3, O_4)^\top$ is used to compute the $p$-value as follows: $P_3 = \Pr(O_3 \leq o_3 \text{ and } O_4 \leq o_4 | H_0) \approx \prod_{i=3}^{4} \Pr(O_i \leq o_i | H_0) = \prod_{i=3}^{4} [\sum_{j \leq o_i} C_j^N (\frac{E_i}{N})^j (1 - \frac{E_i}{N})^{N-j}]$.
- Under Poisson distribution, $O_1$ is used to compute the $p$-value as follows: $P_4 = \Pr(O_1 \geq o_1 | H_0) = \sum_{j \geq o_1} \frac{\exp(-E_1)E_1^j}{j!}$.
- Under Poisson distribution, the test statistic $O_1 + O_2$ is used to compute the $p$-value as follows: $P_5 = P(O_1 + O_2 \geq o_1 + o_2 | H_0) = \sum_{j \geq o_1 + o_2} \frac{\exp[-(E_1+E_2)](E_1+E_2)^j}{j!}$.
- Under Poisson distribution, the test statistic $O_3 + O_4$ is used to compute the $p$-value as follows: $P_6 = P(O_3 + O_4 \leq o_3 + o_4 | H_0) = \sum_{j \leq o_3 + o_4} \frac{\exp[-(E_3+E_4)](E_3+E_4)^j}{j!}$.

If the $p$-value is below the significance level, we conclude that the occurrences of species A and species B are positively associated. These six methods of computing $p$-values can be used for testing negative association by changing the directions of the inequalities in $P_1 - P_6$. For example, if $P_1 = P_r(O_1 \leq o_1 | H_0) = \sum_{j \leq o_1} C_j^N (\frac{E_1}{N})^j (1 - \frac{E_1}{N})^{N-j}$ is lower than the significance level then we have enough evidence to support that the occurrences of the two species are negatively associated.

Besides these six approaches to calculating $p$-values, we also consider the probabilistic approach of *Veech (2013)*, which in our simulation study has been used as a benchmark for comparison with the other approaches. The $p$-value corresponding to this probabilistic approach is given by

$$P_7 = P_r(O_1 \geq o_1 | H_0) = \sum_{j \geq o_1} \frac{C_j^N \times C_{N_b - j}^{N-j} \times C_{N_a - j}^{N - N_b}}{C_{N_b}^N \times C_{N_a}^N},$$

where $\max\{0, N_a + N_b - N\} \leq j \leq \min\{N_a, N_b\}$, is used as $p$-value for testing positive association. It is notable that this method is similar to binomial test but not equivalent. The numerator of the equation is the total number of ways that species A and B could be distributed among $N$ cells for a given $N_a$, $N_b$ and $j$. The denominator represents the total number of ways that species A and B can be arranged among $N$ cells without regard for $j$ (*Veech, 2013*). This quantity $P_7$ can be served as $p$-value for testing negative association through the same modification used for $P_1$.

*Griffith, Veech & Marsh (2016)* developed an R package, *cooccur*, to implement *Veech (2013)*'s method. It is highly accessible and handles large datasets with high performance. All computational procedures in this research were implemented in R software. The R source code is provided in the Supplemental Files and the repository at https://github.com/Yamei628/Prob-Cooccur.

## Simulation scenarios

We conduct a simulation study to compare the performance of our methods with the method proposed by *Veech (2013)*. Functions **runifpoint** and **rpoispp** in *spatstat*, an R software package developed by *Baddeley & Turner (2005)*, are used to simulate spatial
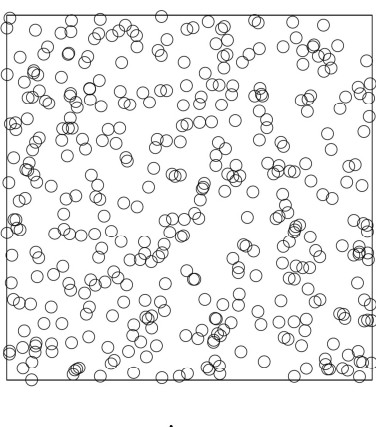 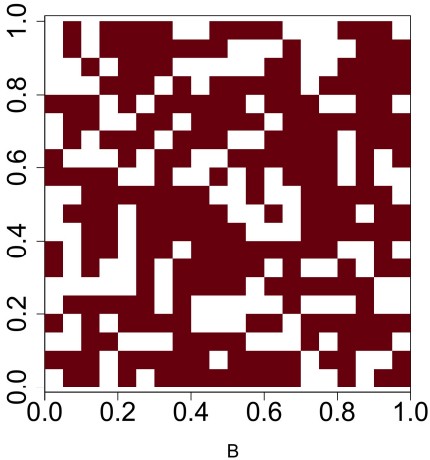

A                                                    B

**Figure 1 Illustration of a uniform point pattern and its presence-absence map.** (A) Randomly distribute 400 points in a $1 \times 1$ plot (B) convert the point pattern process into a presence-absence map where the grid size is $0.05 \times 0.05$.

point patterns. The spatial point patterns are simulated in a $1 \times 1$ plot. The plot is divided into a regular grid where the cell size is $0.05 \times 0.05$ and the number of cells is 400. Function **lets.presab.points** in *letsR*, a R software package developed by *Vilela & Villalobos (2015)*, is used to convert spatial point patterns into presence-absence maps. The test for investigating the association between two species is based on the presence-absence map. Figure 1 illustrates an example of a uniform point pattern and its presence-absence map.

The spatial point patterns are simulated under the following five scenarios.

- Scenario 1: The occurrences of species A and species B are independent. The abundances of these two species are the same and given at various values, $n_a = n_b = 20, 40, 100, 200, 300, \ldots, 1000$.
- Scenario 2: The occurrences of species A and species B are positively associated. The intensities of these two species are the same and given in various settings. The average presence cells of these two species are similar.
- Scenario 3: The occurrences of species A and species B are positively associated. The intensity of species A is fixed, but the intensity of species B is given in various settings. The average presence cells of these two species are dissimilar.
- Scenario 4: The occurrences of species A and species B are negatively associated. The intensities of these two species are both given in various settings. The average presence cells of these two species are similar.
- Scenario 5: The occurrences of species A and species B are negatively associated. The intensity of species A is fixed, but the intensity of species B is given in various settings. The average presence cells of these two species are dissimilar.

For Scenario 1, the function **runifpoint** is used to simulate two random point patterns of given abundances. The Chi-squared statistic Eq. (1) (or Eq. (2) when the Yates' correction is applicable is used to determine whether these two species are independent. For the rest four

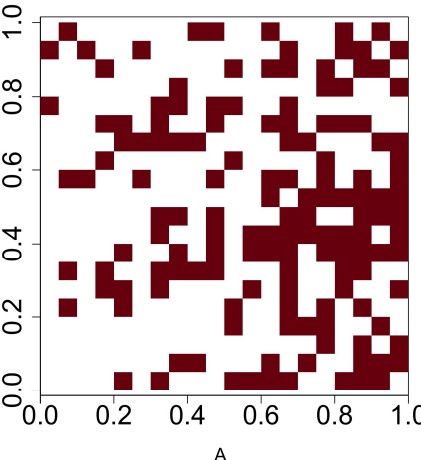 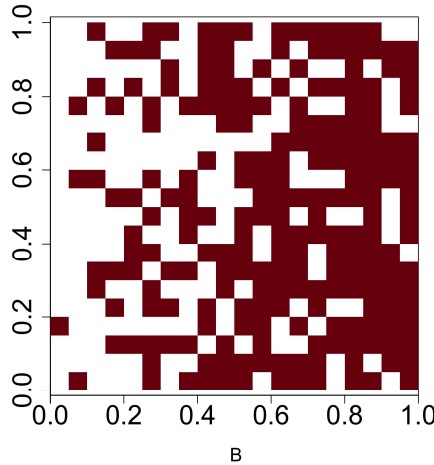

**Figure 2  Presence-absence maps of species A and species B where the intensities of these two species are $\lambda_a(x,y) = 400x$ and $\lambda_b(x,y) = 800x$ respectively.** The occurences of species A and species B are positively associated.

scenarios, the intensities are given and the occurrences of two species are either positively or negatively associated. Function **rpoispp** is used to generate point patterns under these scenarios. Let $\lambda_a(x,y)$ and $\lambda_b(x,y)$ be the intensity function of species A and species B, respectively, where $(x,y)$ are the spatial coordinates of any location in $(0,1)^2 \subset \mathbf{R}^2$. Figure 2 shows the presence-absence maps of species A and species B where $\lambda_a(x,y) = 400x$ and $\lambda_b(x,y) = 800x$. This figure demonstrates an example that the occurrences of two species are positively associated. In contrast, Fig. 3 shows an example that the occurrences of two species are negatively associated, where $\lambda_a(x,y) = 400x$ and $\lambda_b(x,y) = 400 - 400x$.

The level of significance is set at 0.05. For evaluating the proposed methods, the proportion of rejecting the null hypothesis is evaluated based on 500 simulations generated using the true underlying intensity functions.

# RESULTS

## Simulation results

Table 1 presents the simulation results of the Chi-squared test of random association (*i.e.,* the null hypothesis states that the two point patterns are independent of each other) corresponding to Scenario 1 based on the test statistic defined in Eq. (2). The first two columns of the table present the abundance values $n_a$ and $n_b$ considered for two species $A$ and $B$, respectively. The next two columns present the average number of cells ($\bar{N}_a$ and $\bar{N}_b$) that are associated with species presence, and the last column presents the estimated type-I error rate based on 500 simulations.

When the abundance is between 200 and 700, the presence rate (the proportion of the number of presence cells to the number of the overall cells) is between 39.3% and 82.5%. It is an ideal range to use the Chi-squared statistic for testing independence since the estimated type-I error rates are close to the nominal significance level of 5%. When
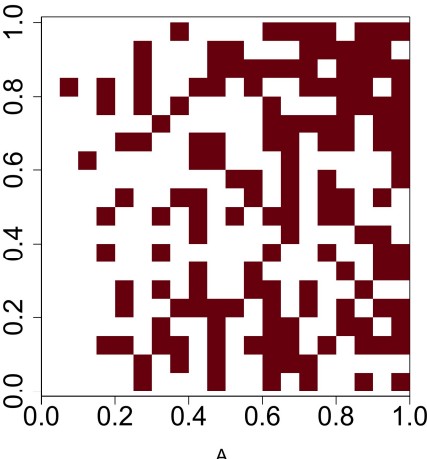
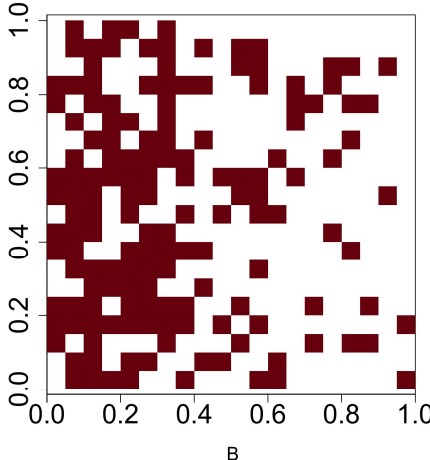

**Figure 3** **Presence-absence maps of species A and species B where the intensities of these two species are $\lambda_a(x,y) = 400x$ and $\lambda_b(x,y) = 400 - 400x$ respectively.** The occurrences of species A and species B are negatively associated.

**Table 1** **Simulation results for Scenario 1.** The abundances of two species are the same and given in the first two columns. The occurrences of two species are independent and simulated in a $1 \times 1$ plot. The plot is divided into a regular grid where the cell size is $0.05 \times 0.05$, and the number of cells is 400. The average presence cells and the type-I error rates of the Chi-squared test are evaluated over 500 simulations.

| Species abundance | | Average pres. cells | | Proportion of rejecting $H_0$ type-I error rate |
|---|---|---|---|---|
| $n_a$ | $n_b$ | $\bar{N}_a$ | $\bar{N}_b$ | |
| 20 | 20 | 19 | 19 | 0.020 |
| 40 | 40 | 39 | 39 | 0.024 |
| 100 | 100 | 89 | 89 | 0.040 |
| 200 | 200 | 158 | 157 | 0.054 |
| 300 | 300 | 211 | 211 | 0.056 |
| 400 | 400 | 253 | 253 | 0.040 |
| 500 | 500 | 285 | 286 | 0.050 |
| 600 | 600 | 311 | 310 | 0.050 |
| 700 | 700 | 330 | 330 | 0.056 |
| 800 | 800 | 346 | 346 | 0.036 |
| 900 | 900 | 358 | 358 | 0.022 |
| 1000 | 1000 | 367 | 367 | 0.018 |

the abundances are outside the range of 200 to 700, the estimated type-I error rates are lower than the nominal significance level. These low estimated type-I error rates indicate that the Chi-squared test of independence may produce high type-II error rates (thus, low statistical powers) when the two species are not independent and have presence rates outside the range of 40% to 80%.

The simulation results of Scenario 2–5 are listed in Tables 2, 3, 4 and 5, respectively. The intensities $\lambda_a(x,y)$ and $\lambda_b(x,y)$ are listed in the first two columns of the tables when the two

**Table 2 Scenario 2 results: intensities used in generating locations of two species for 500 simulations, average abundances, average presences, and the estimated powers $1 - \beta_i$ (highest values in bold) corresponding to the tests based on $P_i$ for $i = 1, \ldots, 7$.**

| Intensity | Average abundance | | Average pres. cells | | Proportion of rejecting $H_0$ (power of the test) | | | | | | |
|---|---|---|---|---|---|---|---|---|---|---|---|
| $\lambda_a, \lambda_b$ | $\bar{n}_a$ | $\bar{n}_b$ | $\bar{N}_a$ | $\bar{N}_b$ | $1 - \beta_1$ | $1 - \beta_2$ | $1 - \beta_3$ | $1 - \beta_4$ | $1 - \beta_5$ | $1 - \beta_6$ | $1 - \beta_7$ |
| $100x$ | 50 | 50 | 46 | 46 | 0.102 | **0.254** | 0.064 | 0.098 | 0.000 | 0.006 | 0.154 |
| $200x$ | 100 | 101 | 85 | 86 | 0.254 | **0.552** | 0.386 | 0.232 | 0.012 | 0.128 | 0.386 |
| $300x$ | 150 | 151 | 118 | 119 | 0.426 | **0.784** | 0.728 | 0.398 | 0.236 | 0.382 | 0.674 |
| $400x$ | 200 | 199 | 148 | 147 | 0.566 | **0.928** | 0.912 | 0.482 | 0.520 | 0.612 | 0.856 |
| $500x$ | 250 | 250 | 172 | 172 | 0.736 | **0.982** | 0.980 | 0.624 | 0.832 | 0.858 | 0.958 |
| $600x$ | 301 | 301 | 193 | 193 | 0.826 | **0.990** | **0.990** | 0.712 | 0.924 | 0.936 | 0.976 |
| $700x$ | 351 | 351 | 212 | 211 | 0.916 | **0.998** | **0.998** | 0.784 | 0.978 | 0.982 | 0.996 |
| $800x$ | 401 | 400 | 227 | 227 | 0.950 | **1.000** | **1.000** | 0.818 | 0.992 | 0.992 | **1.000** |
| $900x$ | 450 | 451 | 241 | 241 | 0.952 | **1.000** | **1.000** | 0.770 | 0.996 | **1.000** | **1.000** |
| $1000x$ | 500 | 498 | 253 | 253 | 0.962 | **1.000** | **1.000** | 0.788 | **1.000** | **1.000** | **1.000** |
| $2000x$ | 1000 | 999 | 321 | 320 | 0.980 | **1.000** | **1.000** | 0.198 | 0.996 | **1.000** | **1.000** |
| $3000x$ | 1501 | 1502 | 347 | 347 | 0.928 | **1.000** | **1.000** | 0.000 | 0.906 | **1.000** | **1.000** |
| $4000x$ | 2002 | 2002 | 361 | 360 | 0.800 | **1.000** | **1.000** | 0.000 | 0.504 | **1.000** | **1.000** |
| $5000x$ | 2500 | 2501 | 368 | 368 | 0.594 | **1.000** | **1.000** | 0.000 | 0.122 | 0.990 | **1.000** |

intensities are different; otherwise, the common intensity is presented in the first column (*e.g.*, in Table 2). In the last seven columns of Tables 2–5, we report the proportions of null hypothesis rejection corresponding to the seven approaches described in the previous section. Note that for Scenario 2–5, these proportions are estimated statistical powers corresponding to the seven approaches. For the $i$th approach ($i = 1, \ldots, 7$), we have used the notation $1 - \beta_i$ to denote the corresponding statistical power, where $\beta_i$ is the estimated type-II error rate.

Tables 2–5 show that the estimated powers of the approaches that use binomial distributions to compute $p$-values are usually higher than those that use Poisson distributions. In particular, the results show that $1 - \beta_1$ is higher than $1 - \beta_4$ for all the scenarios reported in Tables 2–5. The two approaches use the same random variable $O_1$ to compute the $p$-values; however, $1 - \beta_1$ is computed assuming a binomial distribution, whereas $1 - \beta_4$ is computed assuming a Poisson distribution. The second and third approaches, both of which used two variables to compute $p$-values, performed better than the first approach. Overall, the second approach performed best and produced the highest estimated power in most scenarios.

In the scenarios with positive associations (*i.e.*, Scenario 2 and 3), the approaches based on $P_2$, $P_3$, and $P_7$ performed equally well when the average abundances were large for both species. For example, in Scenario 2, the estimated powers of the three approaches are unity for all the cases with large abundances. However, the second approach performed better than the other two in the cases with low abundances. When the average abundance is 50, Table 2 shows that $1 - \beta_2$ is 0.254, which is higher than $1 - \beta_3$ (0.064) and $1 - \beta_7$ (0.154). Similarly, when the average abundance $\bar{n}_b$ is 50 in Table 3, the estimated power of the

**Table 3** Scenario 3 results: intensities used in generating locations of two species for 500 simulations, average abundances, average presences, and the estimated powers $1 - \beta_i$ (highest values in bold) corresponding to the tests based on $P_i$ for $i = 1, \ldots, 7$.

| Intensity | | Average abundance | | Average pres. cells | | Proportion of rejecting $H_0$ (power of the test) | | | | | | |
|---|---|---|---|---|---|---|---|---|---|---|---|---|
| $\lambda_a$ | $\lambda_b$ | $\bar{n}_a$ | $\bar{n}_b$ | $\bar{N}_a$ | $\bar{N}_b$ | $1-\beta_1$ | $1-\beta_2$ | $1-\beta_3$ | $1-\beta_4$ | $1-\beta_5$ | $1-\beta_6$ | $1-\beta_7$ |
| | 100x | | 50 | | 46 | 0.164 | **0.452** | 0.372 | 0.150 | 0.000 | 0.034 | 0.362 |
| | 200x | | 100 | | 85 | 0.336 | **0.712** | 0.656 | 0.306 | 0.126 | 0.244 | 0.602 |
| | 300x | | 150 | | 118 | 0.512 | **0.876** | 0.848 | 0.454 | 0.390 | 0.502 | 0.786 |
| | 400x | | 199 | | 147 | 0.566 | **0.928** | 0.912 | 0.482 | 0.520 | 0.612 | 0.856 |
| | 500x | | 249 | | 171 | 0.710 | **0.956** | 0.952 | 0.612 | 0.736 | 0.782 | 0.910 |
| | 600x | | 299 | | 192 | 0.712 | **0.972** | **0.972** | 0.614 | 0.786 | 0.810 | 0.948 |
| 400x | 700x | 200 | 350 | 148 | 211 | 0.736 | **0.974** | **0.974** | 0.598 | 0.856 | 0.860 | 0.962 |
| | 800x | | 399 | | 227 | 0.772 | 0.992 | **0.994** | 0.608 | 0.910 | 0.902 | 0.982 |
| | 900x | | 449 | | 240 | 0.800 | 0.988 | **0.990** | 0.632 | 0.920 | 0.908 | 0.968 |
| | 1000x | | 499 | | 253 | 0.798 | **0.998** | **0.998** | 0.642 | 0.940 | 0.924 | 0.994 |
| | 2000x | | 1000 | | 320 | 0.588 | **1.000** | **1.000** | 0.228 | 0.956 | 0.858 | **1.000** |
| | 3000x | | 1499 | | 347 | 0.172 | **1.000** | **1.000** | 0.008 | 0.864 | 0.620 | **1.000** |
| | 4000x | | 1999 | | 360 | 0.014 | **0.998** | **0.998** | 0.000 | 0.546 | 0.198 | **0.998** |
| | 5000x | | 2499 | | 369 | 0.000 | 0.986 | **0.994** | 0.000 | 0.184 | 0.028 | 0.990 |

**Table 4** Scenario 4 results: intensities used in generating locations of two species for 500 simulations, average abundances, average presences, and the estimated powers $1 - \beta_i$ (highest values in bold) corresponding to the tests based on $P_i$ for $i = 1, \ldots, 7$.

| Intensity | | Average abundance | | Average pres. cells | | Proportion of rejecting $H_0$ (power of the test) | | | | | | |
|---|---|---|---|---|---|---|---|---|---|---|---|---|
| $\lambda_a$ | $\lambda_b$ | $\bar{n}_a$ | $\bar{n}_b$ | $\bar{N}_a$ | $\bar{N}_b$ | $1-\beta_1$ | $1-\beta_2$ | $1-\beta_3$ | $1-\beta_4$ | $1-\beta_5$ | $1-\beta_6$ | $1-\beta_7$ |
| $100(1-x)$ | 100x | 50 | 50 | 46 | 46 | 0.082 | **0.224** | 0.020 | 0.078 | 0.000 | 0.000 | 0.114 |
| $200(1-x)$ | 200x | 100 | 101 | 85 | 86 | 0.262 | **0.532** | 0.338 | 0.250 | 0.002 | 0.056 | 0.370 |
| $300(1-x)$ | 300x | 150 | 151 | 119 | 119 | 0.448 | **0.756** | 0.712 | 0.394 | 0.182 | 0.304 | 0.664 |
| $400(1-x)$ | 400x | 200 | 199 | 147 | 147 | 0.634 | **0.932** | 0.926 | 0.570 | 0.572 | 0.622 | 0.870 |
| $500(1-x)$ | 500x | 250 | 250 | 172 | 172 | 0.708 | 0.954 | **0.956** | 0.610 | 0.794 | 0.790 | 0.918 |
| $600(1-x)$ | 600x | 300 | 301 | 193 | 193 | 0.768 | **0.990** | **0.990** | 0.654 | 0.884 | 0.870 | 0.966 |
| $700(1-x)$ | 700x | 351 | 351 | 212 | 211 | 0.824 | **0.994** | **0.994** | 0.686 | 0.940 | 0.936 | 0.986 |
| $800(1-x)$ | 800x | 399 | 400 | 227 | 227 | 0.842 | **1.000** | **1.000** | 0.604 | 0.958 | 0.954 | 0.998 |
| $900(1-x)$ | 900x | 450 | 451 | 241 | 241 | 0.872 | **0.998** | **0.998** | 0.598 | 0.966 | 0.972 | 0.996 |
| $1000(1-x)$ | 1000x | 499 | 498 | 253 | 253 | 0.848 | **1.000** | **1.000** | 0.492 | 0.974 | 0.978 | 0.998 |
| $2000(1-x)$ | 2000x | 1000 | 999 | 321 | 320 | 0.036 | **1.000** | **1.000** | 0.000 | 0.404 | 0.966 | **1.000** |
| $3000(1-x)$ | 3000x | 1499 | 1502 | 348 | 347 | 0.000 | **1.000** | 0.964 | 0.000 | 0.000 | 0.048 | 0.998 |
| $4000(1-x)$ | 4000x | 1999 | 2002 | 360 | 360 | 0.000 | **0.996** | 0.062 | 0.000 | 0.000 | 0.000 | 0.970 |
| $5000(1-x)$ | 5000x | 2499 | 2501 | 369 | 368 | 0.000 | **0.784** | 0.000 | 0.000 | 0.000 | 0.000 | 0.222 |

second approach is 0.452, which is higher than 0.372 and 0.362, estimated powers of the third and seventh approaches, respectively.

In the scenarios with negative associations (*i.e.,* Scenario 4 and 5), the top three performers with high estimated powers are the approaches based on $P_2$, $P_3$, and $P_7$. These

Table 5 Scenario 5 results: intensities used in generating locations of two species for 500 simulations, average abundances, average presences, and the estimated powers $1 - \beta_i$ (highest values in bold) corresponding to the tests based on $P_i$ for $i = 1, \ldots, 7$.

| Intensity | | Average abundance | | Average pres. cells | | Proportion of rejecting $H_0$ (power of the test) | | | | | | |
|---|---|---|---|---|---|---|---|---|---|---|---|---|
| $\lambda_a$ | $\lambda_b$ | $\bar{n}_a$ | $\bar{n}_b$ | $\bar{N}_a$ | $\bar{N}_b$ | $1-\beta_1$ | $1-\beta_2$ | $1-\beta_3$ | $1-\beta_4$ | $1-\beta_5$ | $1-\beta_6$ | $1-\beta_7$ |
| | $100x$ | | 50 | | 46 | 0.214 | **0.496** | 0.372 | 0.204 | 0.002 | 0.014 | 0.384 |
| | $200x$ | | 100 | | 85 | 0.390 | **0.738** | 0.682 | 0.364 | 0.114 | 0.202 | 0.632 |
| | $300x$ | | 150 | | 118 | 0.502 | **0.854** | 0.828 | 0.456 | 0.312 | 0.418 | 0.752 |
| | $400x$ | | 199 | | 147 | 0.634 | **0.932** | 0.926 | 0.570 | 0.572 | 0.622 | 0.870 |
| | $500x$ | | 249 | | 171 | 0.638 | **0.938** | **0.938** | 0.560 | 0.658 | 0.670 | 0.894 |
| | $600x$ | | 299 | | 192 | 0.688 | 0.966 | **0.968** | 0.576 | 0.764 | 0.756 | 0.918 |
| $400(1-x)$ | $700x$ | 200 | 350 | 147 | 211 | 0.692 | **0.976** | **0.976** | 0.596 | 0.804 | 0.768 | 0.944 |
| | $800x$ | | 399 | | 227 | 0.700 | **0.968** | **0.968** | 0.578 | 0.824 | 0.782 | 0.942 |
| | $900x$ | | 449 | | 240 | 0.684 | **0.982** | **0.982** | 0.548 | 0.838 | 0.790 | 0.962 |
| | $1000x$ | | 499 | | 253 | 0.704 | 0.984 | **0.992** | 0.562 | 0.880 | 0.816 | 0.966 |
| | $2000x$ | | 1000 | | 320 | 0.376 | 0.976 | **0.984** | 0.168 | 0.786 | 0.602 | 0.960 |
| | $3000x$ | | 1499 | | 347 | 0.100 | 0.930 | **0.954** | 0.012 | 0.558 | 0.284 | 0.922 |
| | $4000x$ | | 1999 | | 360 | 0.018 | 0.882 | **0.928** | 0.000 | 0.286 | 0.072 | 0.886 |
| | $5000x$ | | 2499 | | 369 | 0.000 | 0.802 | **0.884** | 0.000 | 0.140 | 0.024 | 0.810 |

three approaches consistently outperformed the other approaches for the cases reported in Tables 4–5. For Scenario 4, the second approach performed best, especially in the extreme cases where there are very low abundances or very high abundances. For example, when the average abundance is considered to be 50, the estimated power based on $P_2$ is 0.224, which is higher than the estimated powers 0.02 and 0.114 based on $P_3$ and $P_7$, respectively. Moreover, when the average abundance is high (approximately 2,500), the estimated power based on $P_2$ equal to 0.784 (in the last row of Table 4), which is significantly higher than the estimated powers 0.000 and 0.222 based on $P_3$ and $P_7$, respectively. For Scenario 5, when large abundance values are considered for species $B$, the third approach performed slightly better than the second and seventh approaches. For example, in Table 5, the estimated power $1 - \beta_3$ is the highest when the species abundance $\bar{n}_b$ is between 500 and 2500. As with the other scenarios, the second approach performed best for Scenario 5 with low abundance values.

We investigated the type-I error rates of the seven approaches using the nominal significance level 0.05. Table 6 shows the estimated type-I error rates, each calculated based on 500 null samples simulated using Scenario 1. The first two rows and the last row of the table shows that the estimated type-I error rates of the second approach (based on $P_2$) for testing against positive associations are close to the nominal level of 5%. Thus, the second approach will be the most powerful for testing positive association between two species when both presence rates are either below 22.5% or above 90%. When the presence rates are between 22.5% and 90% (*i.e.*, average presence cells are between 90 and 360), the use of the second approach may lead to inflated type-I error rates. In this scenario, we recommend to use either the third or the seventh approach. In the case of assessing negative associations, the estimated type-I error rates of the second approach are close to

the nominal level when the presence rates of both species exceed 85% (*i.e.,* presence cells are more than 340). Because estimated powers of the second approach for these presence rates are highest among the approaches considered in this article, we recommend using the second approach to assess negative associations in the case of large presence rates. Although the estimated powers are high for the second approaches when the presence rates are between 22.5% and 85%, the type-I error estimates for this approach are higher than the nominal level of 5%. In contrast, the type-I error estimates of the seventh approach are below the nominal level. Thus, we recommend using the seventh approach for assessing negative associations when both presence rates are between 22.5% and 85%.

## Data examples
### BCI data analysis

We analyse the popular tropical tree data from Barro Colorado Island (BCI), Panama. The eight most abundant tropical tree species are selected. Our objective is to investigate all 28 pairwise co-occurrence patterns among these eight species. The dataset corresponding to each species is originally stored as a point pattern where each point corresponds to a tree location within the study region. For investigating the co-occurrence pattern of any two given species, we convert the corresponding point patterns into two 5 m $\times$ 5 m gridded presence-absence maps, which allowed us to compute important presence/absence rates for performing statistical tests based on the *p*-values $P_1 - P_7$. Figure 4 shows the 5m $\times$ 5m gridded presence-absence maps of these eight tree species.

Table 7 provides the names and identifiers of the eight species from BCI, along with some key summary measures of them, namely, the total number of trees within the study region (abundance), the total number of cells that are presences (presence cells), and the presence rate (computed as the percentage of total number of cells). For performing pairwise comparisons, we first use the Chi-squared test to test the null hypothesis of independence, and then, if the null hypothesis is rejected, we use the *p*-values $P_i$, i = 1 ,..., 7, to determine whether there is any significant association between two species. Table 8 shows the results of the statistical tests for alternative hypotheses of positive and negative associations. This table also reported the *p*-values corresponding to the Chi-squared test of independence.

Table 8 shows that the Chi-squared test of independence is rejected for all but three pairs: (i) *Faramea occidentalis* and *Alseis blackiana* (No. 2 and 5), (ii) *Trichilia tuberculata* and *Alseis blackiana* (No. 4 and 5), and (iii) *Trichilia tuberculata* and *Mouriri myrtilloides* (No. 4 and 7). Moreover, the results indicate that the six out of eight species (No. 1, 2, 3, 4, 7, and 8) are positively associated; all the pairwise comparisons except the one between the species No. 4 and 7 have produced statistically significant *p*-values at the 5% significance level. The positive association between the species *Hybanthus prunifolius* (No. 1) and *Swartzia simplex* (No. 8) is visible in Fig. 4, as they are nearly absent in the middle-left region. The species *Alseis blackiana* (No. 5) also exhibits positive associations with the four (No. 1, 3, 7, and 8) out of six species in the above mentioned group.

From Fig. 4, we observe that species No. 1 distributes densely on both left-hand and right-hand sides. Species No. 4 and 7 only aggregates on either of the two sides.

Peer∫

**Table 6** Scenario 1 simulation results: species abundances, average presence cells, and the type-I error rates $\alpha_1-\alpha_7$ corresponding to the approaches $P_1-P_7$, respectively, based on 500 simulations from the null hypothesis for testing against the alternative hypotheses of positive and negative associations.

| Species abundance | | Average pres. cells | | $H_1$: positive association | | | | | | | $H_1$: negative association | | | | | | |
|---|---|---|---|---|---|---|---|---|---|---|---|---|---|---|---|---|---|
| $n_a$ | $n_b$ | $\bar{N}_a$ | $\bar{N}_b$ | $\alpha_1$ | $\alpha_2$ | $\alpha_3$ | $\alpha_4$ | $\alpha_5$ | $\alpha_6$ | $\alpha_7$ | $\alpha_1$ | $\alpha_2$ | $\alpha_3$ | $\alpha_4$ | $\alpha_5$ | $\alpha_6$ | $\alpha_7$ |
| 20 | 20 | 19 | 19 | 0.020 | 0.054 | 0.006 | 0.020 | 0.000 | 0.000 | 0.022 | 0.000 | 0.000 | 0.000 | 0.000 | 0.000 | 0.000 | 0.000 |
| 40 | 40 | 39 | 39 | 0.024 | 0.064 | 0.006 | 0.024 | 0.000 | 0.000 | 0.028 | 0.036 | 0.076 | 0.000 | 0.036 | 0.000 | 0.000 | 0.036 |
| 100 | 100 | 89 | 89 | 0.010 | 0.072 | 0.032 | 0.008 | 0.000 | 0.002 | 0.032 | 0.012 | 0.094 | 0.026 | 0.010 | 0.000 | 0.000 | 0.032 |
| 200 | 200 | 158 | 157 | 0.002 | 0.078 | 0.066 | 0.000 | 0.004 | 0.008 | 0.048 | 0.008 | 0.078 | 0.074 | 0.008 | 0.010 | 0.012 | 0.034 |
| 300 | 300 | 211 | 211 | 0.002 | 0.092 | 0.092 | 0.000 | 0.008 | 0.010 | 0.050 | 0.000 | 0.064 | 0.066 | 0.000 | 0.012 | 0.012 | 0.034 |
| 400 | 400 | 253 | 253 | 0.000 | 0.078 | 0.066 | 0.000 | 0.002 | 0.004 | 0.038 | 0.000 | 0.074 | 0.064 | 0.000 | 0.000 | 0.002 | 0.032 |
| 500 | 500 | 285 | 286 | 0.000 | 0.078 | 0.048 | 0.000 | 0.000 | 0.004 | 0.032 | 0.000 | 0.076 | 0.056 | 0.000 | 0.002 | 0.002 | 0.040 |
| 600 | 600 | 311 | 310 | 0.000 | 0.088 | 0.050 | 0.000 | 0.000 | 0.004 | 0.052 | 0.000 | 0.086 | 0.040 | 0.000 | 0.000 | 0.002 | 0.040 |
| 700 | 700 | 330 | 330 | 0.000 | 0.076 | 0.036 | 0.000 | 0.000 | 0.002 | 0.042 | 0.000 | 0.082 | 0.016 | 0.000 | 0.000 | 0.000 | 0.038 |
| 800 | 800 | 346 | 346 | 0.000 | 0.106 | 0.026 | 0.000 | 0.000 | 0.002 | 0.058 | 0.000 | 0.060 | 0.004 | 0.000 | 0.000 | 0.000 | 0.024 |
| 900 | 900 | 358 | 358 | 0.000 | 0.082 | 0.004 | 0.000 | 0.000 | 0.002 | 0.026 | 0.000 | 0.052 | 0.000 | 0.000 | 0.000 | 0.000 | 0.018 |
| 1000 | 1000 | 367 | 367 | 0.000 | 0.052 | 0.008 | 0.000 | 0.000 | 0.000 | 0.024 | 0.000 | 0.048 | 0.000 | 0.000 | 0.000 | 0.000 | 0.010 |

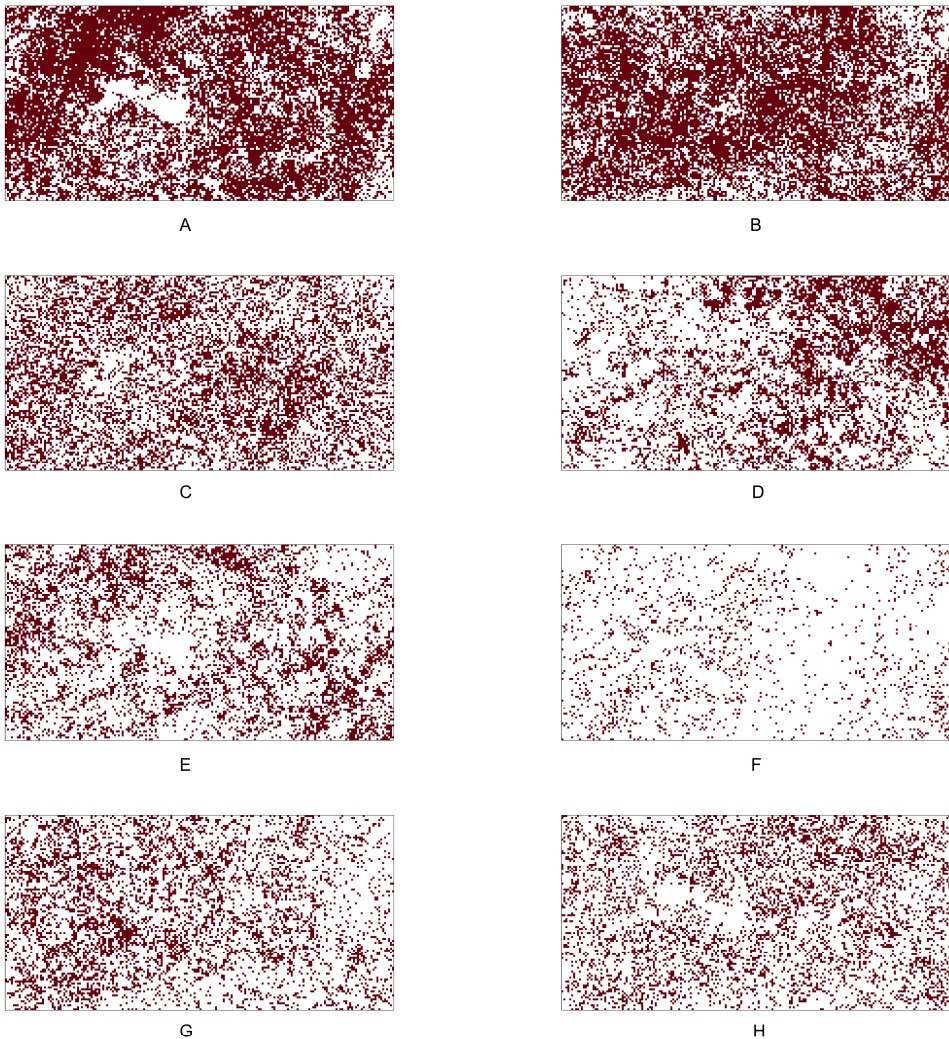

**Figure 4** **Presence-absence maps of the eight most abundant species in BCI tree data.** (A) Hybanthus prunifolius. (B) Faramea occidentalis.(C) Desmopsis panamensis. (D) Trichilia tuberculata. (E) Alseis blackiana. (F) Oenocarpus mapora. (G) Mouriri myrtilloides (H) Swartzia simplex. The area is 50-ha ($500 \times 1000 \ m^2$). The corresponding presence-absence maps are at cell size $5 \times 5$ m.

Consequently, the testing results show No. 1 is positively associated with species No. 4 and 7, but the latter two species are not positively associated. This example demonstrates that we can not guarantee the positive association between two species if they are positively associated with the same species.

Table 7 reveals that the species *Oenocarpus mapora* (No. 6) possesses a unique feature of a high abundance but low presence. This may be evident from the spatial distribution displayed in Fig. 4F. While it exhibits clustering at the local scale, there appear to be large distances between clusters at the global scale (*i.e.,* within the entire study region). The results in Table 8 show that *Oenocarpus mapora* is negatively associated with all the other species.

**Table 7 Eight most abundant species in BCI tree data.** The data contains tree locations and the study area is 50-ha ($500 \times 1000$ m²). The data are converted into $5 \times 5$ *m* gridded presence-absence maps. The presence cells and the presence rates are computed from the presence-absence maps.

| No. | Species | Abundance | Presence cells | Presence rate |
|---|---|---|---|---|
| 1 | *Hybanthus prunifolius* | 37123 | 12053 | 60.3% |
| 2 | *Faramea occidentalis* | 26420 | 12060 | 60.3% |
| 3 | *Desmopsis panamensis* | 13048 | 7901 | 39.5% |
| 4 | *Trichilia tuberculata* | 11779 | 6521 | 32.6% |
| 5 | *Alseis blackiana* | 8700 | 5737 | 28.7% |
| 6 | *Oenocarpus mapora* | 7947 | 1623 | 8.1% |
| 7 | *Mouriri myrtilloides* | 7538 | 5305 | 26.5% |
| 8 | *Swartzia simplex* | 6103 | 4991 | 25.0% |

### Lansing data analysis

Table 9 shows the identification number (No.), species name, total number of tree locations, total number of presence cells, and presence rate for the six tree species in the Lansing dataset. Figure 5 shows the $0.05 \times 0.05$ gridded presence-absence maps. We examine all 15 pairwise co-occurrence patterns that are possible among the six species.

The results of the Chi-squared tests, the tests of positive associations, and the tests of negative associations are presented in Table 10. The results of the Chi-squared test in the third column show that the null hypothesis of random association can be rejected for six pairs at the 5% significance level. These pairs are: (i) black oak and maple (*p*-value: 0.015); (ii) black oak and miscellaneous (*p*-value $< 0.001$); (iii) hickory and maple (*p*-value $< 0.001$); (iv) hickory and miscellaneous (*p*-value: 0.017); (v) hickory and white oak(*p*-value: 0.006); and (vi) maple and miscellaneous (*p*-value $< 0.001$).

Based on most of the *p*-values reported in Table 10, we conclude that, except for the pair maple and miscellaneous, all other pairs mentioned above are negatively associated. Note that all the *p*-values (shown in columns 4–10 in Table 10) obtained from testing the positive co-occurrence between maple and miscellaneous are smaller than the standard significance level of 0.05, indicating that this pair has a strong positive co-occurrence pattern. The presence-absence maps in Fig. 5 can also be used to informally verify that the pair maple and miscellaneous exhibit similar presence-absence patterns within the study window. Both categories are rare in the top-left part of the study window, while they are relatively abundant in the bottom and top-right parts.

Both the species black oak and hickory are negatively associated with the same two species, namely, maple and miscellaneous, one may suspect that they form a positive co-occurrence pattern. However, based on the Chi-squared test result, we can conclude that the species black oak and hickory are randomly co-occurring (*p*-value: 0.22).

Many of the aforementioned negative associations can also be verified by visually comparing the point patterns or presence-absence maps between two species. For example, Figure 5 illustrates that, while both maple and miscellaneous are sparse in the upper-left and upper-right corners of the study window, the species black oak and hickory are

Chang et al. (2023), *PeerJ*, DOI 10.7717/peerj.15907

**Table 8  Using eight most abundant species in BCI tree data for testing positive association.** The first two columns are the identity numbers (given in Table 7) of two species. The *p*-values by using the Chi-squared statistic to test the null hypothesis of independence are listed in the third column. The rest columns are the *p*-values of $P_i$ for testing positive or negative association, $i = 1, \ldots, 7$.

| Species No. | | Chi-sq. test *p*-value | *p*-values for testing positive association | | | | | | | *p*-values for testing negative association | | | | | | |
|---|---|---|---|---|---|---|---|---|---|---|---|---|---|---|---|---|
| A | B | | $P_1$ | $P_2$ | $P_3$ | $P_4$ | $P_5$ | $P_6$ | $P_7$ | $P_1$ | $P_2$ | $P_3$ | $P_4$ | $P_5$ | $P_6$ | $P_7$ |
| 1 | 2 | **0.000** | 0.000 | 0.000 | 0.000 | 0.003 | 0.000 | 0.000 | 0.000 | 1.000 | 1.000 | 1.000 | 0.997 | 1.000 | 1.000 | 1.000 |
| 1 | 3 | **0.000** | 0.000 | 0.000 | 0.000 | 0.000 | 0.000 | 0.000 | 0.000 | 1.000 | 1.000 | 1.000 | 1.000 | 1.000 | 1.000 | 1.000 |
| 1 | 4 | **0.000** | 0.002 | 0.000 | 0.000 | 0.005 | 0.000 | 0.001 | 0.000 | 0.998 | 0.994 | 0.991 | 0.995 | 1.000 | 0.999 | 1.000 |
| 1 | 5 | **0.000** | 0.000 | 0.000 | 0.000 | 0.000 | 0.000 | 0.000 | 0.000 | 1.000 | 1.000 | 1.000 | 1.000 | 1.000 | 1.000 | 1.000 |
| 1 | 6 | **0.000** | 1.000 | 1.000 | 1.000 | 1.000 | 1.000 | 1.000 | 1.000 | 0.000 | 0.000 | 0.000 | 0.000 | 0.000 | 0.000 | 0.000 |
| 1 | 7 | **0.000** | 0.000 | 0.000 | 0.000 | 0.000 | 0.000 | 0.000 | 0.000 | 1.000 | 1.000 | 1.000 | 1.000 | 1.000 | 1.000 | 1.000 |
| 1 | 8 | **0.002** | 0.037 | 0.003 | 0.002 | 0.050 | 0.028 | 0.041 | 0.001 | 0.965 | 0.889 | 0.890 | 0.952 | 0.973 | 0.959 | 0.999 |
| 2 | 3 | **0.000** | 0.000 | 0.000 | 0.000 | 0.000 | 0.000 | 0.000 | 0.000 | 1.000 | 1.000 | 1.000 | 1.000 | 1.000 | 1.000 | 1.000 |
| 2 | 4 | **0.000** | 0.001 | 0.000 | 0.000 | 0.003 | 0.000 | 0.000 | 0.000 | 0.999 | 0.996 | 0.993 | 0.997 | 1.000 | 1.000 | 1.000 |
| 2 | 5 | 0.242 | 0.250 | 0.071 | 0.064 | 0.269 | 0.223 | 0.243 | 0.124 | 0.756 | 0.545 | 0.558 | 0.736 | 0.780 | 0.760 | 0.882 |
| 2 | 6 | **0.000** | 1.000 | 1.000 | 1.000 | 1.000 | 1.000 | 1.000 | 1.000 | 0.000 | 0.000 | 0.000 | 0.000 | 0.000 | 0.000 | 0.000 |
| 2 | 7 | **0.000** | 0.000 | 0.000 | 0.000 | 0.000 | 0.000 | 0.000 | 0.000 | 1.000 | 1.000 | 1.000 | 1.000 | 1.000 | 1.000 | 1.000 |
| 2 | 8 | **0.000** | 0.000 | 0.000 | 0.000 | 0.000 | 0.000 | 0.000 | 0.000 | 1.000 | 1.000 | 0.999 | 1.000 | 1.000 | 1.000 | 1.000 |
| 3 | 4 | **0.000** | 0.000 | 0.000 | 0.000 | 0.001 | 0.001 | 0.001 | 0.000 | 1.000 | 0.988 | 0.992 | 0.999 | 0.999 | 1.000 | 1.000 |
| 3 | 5 | **0.000** | 0.002 | 0.000 | 0.000 | 0.003 | 0.006 | 0.003 | 0.000 | 0.998 | 0.970 | 0.976 | 0.997 | 0.994 | 0.997 | 1.000 |
| 3 | 6 | **0.000** | 1.000 | 1.000 | 1.000 | 1.000 | 1.000 | 1.000 | 1.000 | 0.000 | 0.000 | 0.000 | 0.000 | 0.000 | 0.000 | 0.000 |
| 3 | 7 | **0.000** | 0.000 | 0.000 | 0.000 | 0.000 | 0.000 | 0.000 | 0.000 | 1.000 | 0.996 | 0.998 | 1.000 | 1.000 | 1.000 | 1.000 |
| 3 | 8 | **0.000** | 0.000 | 0.000 | 0.000 | 0.000 | 0.002 | 0.001 | 0.000 | 1.000 | 0.985 | 0.990 | 1.000 | 0.998 | 0.999 | 1.000 |
| 4 | 5 | 0.376 | 0.744 | 0.483 | 0.463 | 0.733 | 0.691 | 0.720 | 0.816 | 0.264 | 0.094 | 0.106 | 0.274 | 0.312 | 0.284 | 0.193 |
| 4 | 6 | **0.000** | 1.000 | 1.000 | 1.000 | 1.000 | 1.000 | 1.000 | 1.000 | 0.000 | 0.000 | 0.000 | 0.000 | 0.000 | 0.000 | 0.000 |
| 4 | 7 | 0.992 | 0.501 | 0.251 | 0.252 | 0.500 | 0.499 | 0.500 | 0.502 | 0.509 | 0.257 | 0.255 | 0.509 | 0.505 | 0.504 | 0.511 |
| 4 | 8 | **0.024** | 0.049 | 0.009 | 0.016 | 0.056 | 0.117 | 0.078 | 0.013 | 0.953 | 0.783 | 0.765 | 0.946 | 0.885 | 0.924 | 0.988 |
| 5 | 6 | **0.000** | 1.000 | 0.997 | 0.998 | 1.000 | 0.999 | 1.000 | 1.000 | 0.000 | 0.000 | 0.000 | 0.000 | 0.001 | 0.000 | 0.000 |
| 5 | 7 | **0.000** | 0.000 | 0.000 | 0.000 | 0.000 | 0.006 | 0.001 | 0.000 | 1.000 | 0.974 | 0.986 | 1.000 | 0.994 | 0.999 | 1.000 |
| 5 | 8 | **0.000** | 0.004 | 0.000 | 0.002 | 0.006 | 0.039 | 0.014 | 0.000 | 0.996 | 0.914 | 0.921 | 0.995 | 0.962 | 0.986 | 1.000 |
| 6 | 7 | **0.000** | 1.000 | 0.992 | 0.996 | 1.000 | 0.997 | 1.000 | 1.000 | 0.000 | 0.000 | 0.000 | 0.000 | 0.003 | 0.000 | 0.000 |
| 6 | 8 | **0.000** | 1.000 | 0.996 | 0.998 | 1.000 | 0.998 | 1.000 | 1.000 | 0.000 | 0.000 | 0.000 | 0.000 | 0.002 | 0.000 | 0.000 |
| 7 | 8 | **0.000** | 0.003 | 0.000 | 0.001 | 0.004 | 0.038 | 0.012 | 0.000 | 0.997 | 0.919 | 0.929 | 0.997 | 0.963 | 0.989 | 1.000 |

**Notes.**

The bold font indicates the p-value (by using Chi-squared statistic) which is lower than the 5% significance level.

**Table 9  Lansing data contains locations of six tree species. The study plot is 19.6-acre (924 × 924 ft²).** The plot has been re-scaled to the unit square. We converted it into 0.05 × 0.05 gridded presence-absence maps. The presence cells and the presence rates are computed from the presence-absence maps.

| No. | Species | Abundance | Presence cells | Presence rate |
|---|---|---|---|---|
| 1 | black oak | 135 | 93 | 23.3% |
| 2 | hickory | 703 | 285 | 71.3% |
| 3 | maple | 514 | 212 | 53.0% |
| 4 | miscellaneous | 105 | 72 | 18.0% |
| 5 | red oak | 346 | 198 | 49.5% |
| 6 | white oak | 448 | 247 | 61.8% |

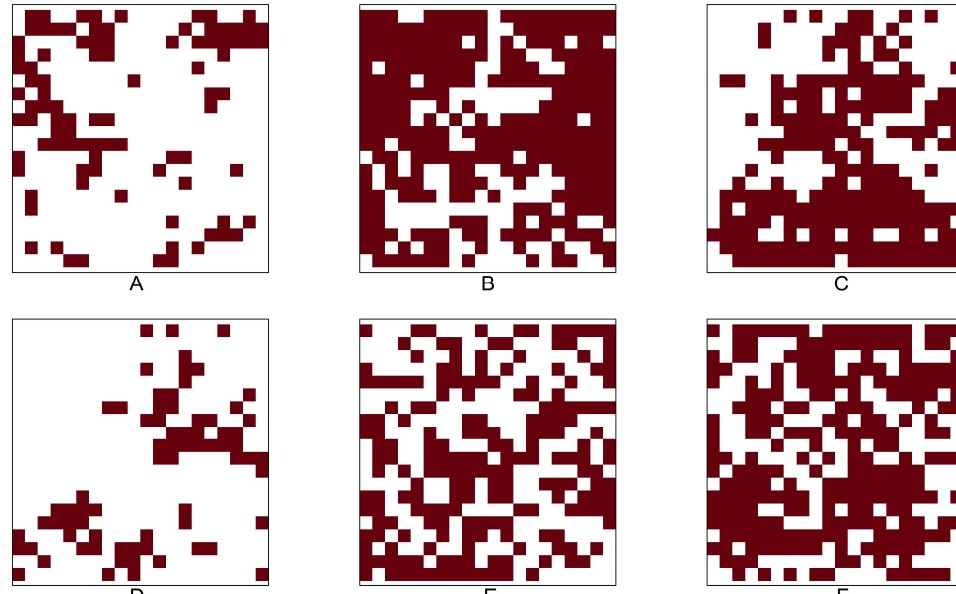

**Figure 5  Presence-absence maps of Lansing data: (A) black oak. (B) hickory. (C) maple. (D) miscellaneous. (E) red oak. (F) white oak.** The study plot has been re-scaled to the unit square. We converted the original data into 0.05 × 0.05 gridded presence-absence maps.

dense in these two corners. The statistical tests become indispensable when other simpler methods, such as visual inspection of point patterns, do not provide a clear picture of the co-occurrence pattern for any given species pair. For example, the patterns of hickory and white oak do not provide any clear indication about their co-occurrence pattern; however, the $p$-values obtained from testing the negative co-occurrence reveal that hickory and white oak might be negatively associated with each other.

## DISCUSSION

The Chi-squared test is simple and intuitive. Its type I error rates are close to the significance level when the presence rates are around 40% ∼ 80%. The simulation results reveal

Chang et al. (2023), *PeerJ*, DOI 10.7717/peerj.15907

**Table 10 Using Lansing data for testing positive association.** The first two columns are the identity numbers (given in Table 9) of two species. The *p*-values by using the Chi-squared statistic to test the null hypothesis of independence are listed in the third column. The rest columns are the *p*-values of $P_i$ for testing positive or negative association, $i = 1, \ldots, 7$.

| Species No. | | Chi-sq. test | *p*-values for testing positive association | | | | | | | *p*-values for testing negative association | | | | | | |
|---|---|---|---|---|---|---|---|---|---|---|---|---|---|---|---|---|
| A | B | *p*-value | $P_1$ | $P_2$ | $P_3$ | $P_4$ | $P_5$ | $P_6$ | $P_7$ | $P_1$ | $P_2$ | $P_3$ | $P_4$ | $P_5$ | $P_6$ | $P_7$ |
| 1 | 2 | 0.215 | 0.281 | 0.085 | 0.067 | 0.296 | 0.233 | 0.286 | 0.133 | 0.762 | 0.562 | 0.598 | 0.744 | 0.790 | 0.736 | 0.916 |
| 1 | 3 | **0.015** | 0.953 | 0.829 | 0.825 | 0.942 | 0.937 | 0.927 | 0.995 | 0.065 | 0.010 | 0.010 | 0.078 | 0.073 | 0.083 | 0.010 |
| 1 | 4 | **0.000** | 1.000 | 0.929 | 0.942 | 1.000 | 0.958 | 0.991 | 1.000 | 0.000 | 0.000 | 0.001 | 0.000 | 0.048 | 0.011 | 0.000 |
| 1 | 5 | 0.472 | 0.705 | 0.452 | 0.459 | 0.693 | 0.675 | 0.683 | 0.799 | 0.352 | 0.140 | 0.135 | 0.362 | 0.351 | 0.343 | 0.274 |
| 1 | 6 | 0.186 | 0.800 | 0.593 | 0.616 | 0.780 | 0.804 | 0.777 | 0.925 | 0.244 | 0.072 | 0.060 | 0.262 | 0.218 | 0.243 | 0.115 |
| 2 | 3 | **0.000** | 0.984 | 0.983 | 0.982 | 0.955 | 0.998 | 0.998 | 1.000 | 0.021 | 0.000 | 0.000 | 0.053 | 0.002 | 0.003 | 0.000 |
| 2 | 4 | **0.017** | 0.908 | 0.772 | 0.791 | 0.893 | 0.924 | 0.858 | 0.993 | 0.120 | 0.022 | 0.010 | 0.137 | 0.089 | 0.156 | 0.014 |
| 2 | 5 | 0.190 | 0.284 | 0.062 | 0.063 | 0.320 | 0.209 | 0.213 | 0.115 | 0.750 | 0.615 | 0.613 | 0.709 | 0.810 | 0.808 | 0.922 |
| 2 | 6 | **0.006** | 0.896 | 0.879 | 0.869 | 0.826 | 0.953 | 0.964 | 0.998 | 0.123 | 0.004 | 0.007 | 0.194 | 0.055 | 0.042 | 0.004 |
| 3 | 4 | **0.000** | 0.014 | 0.001 | 0.000 | 0.019 | 0.027 | 0.028 | 0.000 | 0.990 | 0.920 | 0.924 | 0.987 | 0.977 | 0.977 | 1.000 |
| 3 | 5 | 0.556 | 0.649 | 0.424 | 0.432 | 0.626 | 0.671 | 0.677 | 0.755 | 0.394 | 0.154 | 0.149 | 0.412 | 0.356 | 0.348 | 0.312 |
| 3 | 6 | 0.985 | 0.515 | 0.266 | 0.274 | 0.508 | 0.504 | 0.514 | 0.533 | 0.527 | 0.283 | 0.274 | 0.526 | 0.524 | 0.515 | 0.549 |
| 4 | 5 | 0.868 | 0.570 | 0.311 | 0.322 | 0.565 | 0.545 | 0.555 | 0.616 | 0.500 | 0.248 | 0.237 | 0.502 | 0.483 | 0.473 | 0.486 |
| 4 | 6 | 0.885 | 0.489 | 0.242 | 0.253 | 0.488 | 0.477 | 0.489 | 0.498 | 0.574 | 0.314 | 0.309 | 0.572 | 0.553 | 0.537 | 0.607 |
| 5 | 6 | 0.502 | 0.656 | 0.446 | 0.454 | 0.628 | 0.688 | 0.693 | 0.781 | 0.385 | 0.141 | 0.136 | 0.407 | 0.338 | 0.332 | 0.285 |

**Notes.**

The bold font indicates the p-value (by using Chi-squared statistic) which is lower than the 5% significance level.

that it may suffer high type II error rates for extremely low or high presence rate data. Moreover, when the null hypothesis is rejected, it can not distinguish the positive or negative association between two species. We suggest to use it as a preliminary tool for supporting evidence.

*Veech (2013)*'s method is also easy to implement. It is recommended when the presence rate is around $22.5\% \sim 85\%$. Within this range, its testing powers achieve 90%, and the type I error rates are below or close to the significance level. But outside this range, our second approach ($P_2$ which uses $O_1$ and $O_2$ under binomial distribution) serves as a useful alternative since it not only has the closest type I error rates to the significance level but also outperforms the others in terms of testing powers among most scenarios.

Under the assumption that every individual is randomly distributed in each cell with equal probability, $(O_1 + O_2)$ follows a binomial distribution with $Bin(N, (E_1 + E_2)/N)$ since they are count variables of two mutually exclusive events. We attempted to compute $p$-values by using the sum of $O_1 + O_2$ under this binomial distribution. The results were similar to Veech's method when the presence rate was moderate, but inferior to Veech's method when the presence rate was either low or high. However, our $P_2$ method outperforms Veech's in situations of low or high presence rates. We assumed that $O_1$ and $O_2$ were approximately independent while constructing $P_2$. Although $(O_1, O_2, O_3, O_4)$ follows a multinomial distribution with $Cov(O_i, O_j) = -(E_i * E_j)/N, i \neq j$. When $N$ is large and either $E_1$ or $E_2$ is small, $O_1$ and $O_2$ are approximately uncorrelated. That explains why our $P_2$ method outperforms Veech's when the presence rate is either low (small $E_1$) or high (small $E_2$).

Our methods, as well as Veech's (*Veech, 2013*), solely use information from the two-way contingency table, lacking any spatial context. However, we applied these methods to presence-absence data over a spatial grid as this type of data is commonly encountered in ecology. Both our simulation and application studies were conducted under this setting. Although we have made efforts to incorporate spatial context into our analysis, the corresponding results were disappointing. We are actively working on refining our methods and striving to achieve more satisfactory outcomes. For references that have considered the spatial context in their analysis, please refer to *Rizzo et al. (2023)* and *Sallam et al. (2023)*.

The simulated data were initially generated as spatial point patterns, as it is easier to construct two independent or associated species within this framework. The two real datasets are also originally classified as spatial point patterns since they are readily accessible. To implement our methods and Veech's method (*Veech, 2013*), we converted these spatial point pattern data into presence-absence maps.

The application results of these probabilistic approaches were consistent. If the Chi-squared test was rejected, then our six approaches and *Veech (2013)*'s method all confirmed that there was a significant association between two species. These two data examples illustrate that if two species are positively or negatively associated with the same species, it does not imply the two species are associated. The results show that we can provide statistical evidence to support concepts of existing phenomena or provide additional information.

The proposed methods did not take false absences/presences or missing values into account. *Veech (2013)* referred that false absences/presences simply represent a source of type-I and II errors and the neglect of them probably does not bias the model toward finding positive or negative association. As for missing values, if they are missing at random, it is seasonable to use zero, median, or k-nearest neighbor (KNN) to impute. But missing patterns normally are not randomly distributed. We need to modify these imputation methods to improve its efficiency.

The probabilistic methods discussed in this article did not use spatial statistics approaches. The kernel estimation method (*Diggle, 1985*; *Silverman, 1986*; *Baddeley, Rubak & Turner, 2015*) is useful to estimate the occurrence rate for each cell. The cross-K function (*Baddeley, Rubak & Turner, 2015*) which investigate the dependence between two point patterns can also be considered. How to implement these spatial statistics methods into co-occurrence research also became our research interests.

## ACKNOWLEDGEMENTS

Some results of this article appear in the master theses of Jyun-Hung Huang (*Huang, 2020*) and Wen-Hsuan Wu (*Wu, 2020*). The BCI plot project is part of the Forest Global Earth Observatory (ForestGEO), a global network of large-scale demographic tree plots. We are grateful to ForestGEO for providing valuable datasets. We also express our appreciation to the editor and the reviewers for devoting their time and effort to provide us with valuable comments.

### Funding

This work was supported by the National Science and Technology Council of Taiwan (No. 111-2635-M-032-001 and 107-2118-M-032-007). The BCI forest dynamics research project was made possible by National Science Foundation grants to Stephen P. Hubbell: DEB-0640386, DEB-0425651, DEB-0346488, DEB-0129874, DEB-00753102, DEB-9909347, DEB-9615226, DEB-9615226, DEB-9405933, DEB-9221033, DEB-9100058, DEB-8906869, DEB-8605042, DEB-8206992, DEB-7922197, support from the Forest Global Earth Observatory (ForestGEO), the Smithsonian Tropical Research Institute, the John D. and Catherine T. MacArthur Foundation, the Mellon Foundation, and the Small World Institute Fund. The funders had no role in study design, data collection and analysis, decision to publish, or preparation of the manuscript.

### Grant Disclosures

The following grant information was disclosed by the authors:
National Science and Technology Council of Taiwan: 111-2635-M-032-001, 107-2118-M-032-007.
National Science Foundation: DEB-0640386, DEB-0425651, DEB-0346488, DEB-0129874, DEB-00753102, DEB-9909347, DEB-9615226, DEB-9615226, DEB-9405933, DEB-9221033, DEB-9100058, DEB-8906869, DEB-8605042, DEB-8206992, DEB-7922197.

Forest Global Earth Observatory (ForestGEO).
Smithsonian Tropical Research Institute.
The John D. and Catherine T. MacArthur Foundation, the Mellon Foundation.
The Mellon Foundation.
The Small World Institute Fund.

## Competing Interests

The authors declare there are no competing interests.

## Author Contributions

- Ya-Mei Chang conceived and designed the experiments, performed the experiments, analyzed the data, prepared figures and/or tables, authored or reviewed drafts of the article, and approved the final draft.
- Suman Rakshit analyzed the data, authored or reviewed drafts of the article, and approved the final draft.
- Chun-Hung Huang performed the experiments, analyzed the data, prepared figures and/or tables, and approved the final draft.
- Wen-Hsuan Wu performed the experiments, analyzed the data, prepared figures and/or tables, and approved the final draft.

## Data Availability

The BCI data is available at ForestGEO: https://forestgeo.si.edu. The Lansing data is available from the R package at https://spatstat.org.

The code for analyzing Lansing data is available in the Supplemental File.

The complete code for simulation and application is available at Github and Zenodo: https://github.com/Yamei628/Prob-Cooccur

Yamei628. (2023). Yamei628/Prob-Cooccur: Prob-Cooccur (ProbCooccur.v1.0.0). Zenodo. https://doi.org/10.5281/zenodo.8213296.

## Supplemental Information

Supplemental information for this article can be found online at http://dx.doi.org/10.7717/peerj.15907#supplemental-information.

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
