# Peer review of "Probabilistic approaches for investigating species co-occurrence from presence-absence maps"

_PeerJ, doi:10.7717/peerj.15907_

## Round 0.1 · original submission · Minor Revisions

Dear authors,

Two reviewers have highlighted various improvements that can be made to the paper. Please seriously read their comments and send us a revised version.

Please change your wording on Veech et al and other prior work as suggested by Reviewer 1.

Reviewer 2 called for major revisions, primarily related to the mismatch between what that reviewer calls cases 1, 2, and 3. You could deal with this in two ways. The first, which is much harder, is to add experiments dealing with the spatial context the reviewer finds missing. The second is to revise your wording so it is very clear which of the 3 cases you are addressing with your methods and analyses.

Reviewer 1 ·

Basic reporting

My greatest concern for this paper is that Chang et al. are incorrect in some of their statements about previously proposed (and widely used) tests of species co-occurrence, such as the probabilistic model of Veech (2013, 2014, also Griffith et al. 2016) and the approach of Sfenthourakis et al. (2004, 2006). Chang et al. claim that if two species are not distributed independently of one another then one cannot use these tests. They are wrong about this. Indeed, the Veech/Sfenthorakis methods are actually based on testing whether two species are distributed independently of one another. If a non-significant p-value is obtained then the best (most appropriate) inference is that the distribution of each species does not depend on the distribution of the other species. That is, the two species are distributed independently of one another. If a significant p-value is obtained, then the two species may be positively associated (if observed co-occurrence exceeds expected co-occurrence) or negatively associated (if observed co-occurrence is less than expected co-occurrence). Therefore, there is nothing about the previous methods of Veech/Sfenthourakis that needs to be corrected. The paper of Chang et al. may still have some value in that they do address how species incidence rate affects the performance of co-occurrence tests, although other authors have also explored this issue. Chang et al. also propose ways of obtaining p-values (to assess significantly positive or negative co-occurrence) based on treating a series of four O variables as either binomial or Poisson variables. The O variables represented observed occurrence of Species 1, observed occurrence of Species 2, occurrence of both, and occurrence of neither, wherein occurrence is measured as number of grid cells. Although these “new” p-values may be useful in some circumstances, they seem overly convoluted and make the testing for non-random co-occurrence patterns much more complicated than it needs to be. I do not see that the proposed methods of Chang et al. are any kind of improvement in testing and studying species co-occurrence. Nonetheless, I suppose the paper is publishable, even though the authors have misrepresented previous papers.

Experimental design

no comment

Validity of the findings

The paper misrepresents previously published studies.

Additional comments

none

Reviewer 2 ·

Basic reporting

In their manuscript titled “Probabilistic approaches for investigating species co-occurrence from presence-absence maps” the authors focus on creation of several novel statistical testing procedures aimed to assess the potential existence and sign of species-to-species co-occurrence patterns in spatial species community data. The authors present 6 tests for positive co-occurrence assessment and 6 tests for negative co-occurrence assessment, for which they explicitise the p-value computations. They investigate the behaviour of these tests against each other and against a couple of widely used alternatives using a set of simulation scenarios that differ in simulation settings. Finally, they apply the presented approach to two real datasets on tree distributions.

In my opinion, the manuscript is written with good command of professional English. On the other hand, the narrative and composition of the manuscript can be significantly enhanced further. This is especially relevant to the usage of tables and figures, which number is overwhelmingly large, but the contribution of many seems to be suboptimal. Not being an expert in the very field of pairwise approaches for species co-occurrence pattern assessment, I do not feel competent enough to evaluate whether the manuscript properly acknowledges the key existing literature in this field in a way that would provide enough context and background. The manuscript features an R code supplement that cover only one of real data experiments and, therefore, does not enable replicability assessment of all results presented in the manuscript.

Experimental design

My biggest concerns are in the scope of the match between declared aims (as they appear in the title) and the proposed methods. Thus, the authors deal with three types of data within their own manuscript: 1) point-referenced spatial data on species presence, 2) presence-absence data over a spatial grid and 3) occurrence and co-occurrence counts that lack any spatial context. The manuscript title and introduction section suggest that the study is supposed to be focused on the 2). In contrast, the datasets used are all originally in the 1) form, and the proposed methods seem to be most suitable for the type 3) data. Indeed, in the very end of the Discussion section, the authors briefly address the importance of accounting for spatial dependence, but the lack of this treatment for explicit spatial context in the conducted research puts a critical concern on whether the stress on spatial context in the title is properly addressed in the body of the manuscript.

Validity of the findings

My concern for the lack of treatment of spatial context also expands to the presented conclusions.

Additional comments

Minor comments
L17-18, 69-70. The repeated sentence “We focus on using presence-absence maps on a certain study plot instead.” is not clear to me. Do the authors imply that the proposed methods are not suitable for spatially non-continuous data?

L99-101, 122. It is not clear, what the author label as the “null hypothesis of randomness” in this place. Assuming that it is same as “null hypothesis” in Chi-squared test - so that the distribution of considered species is independent from each other, it is unclear of how the authors reach the conclusion that each random variable O_i follows the binomial distribution, given that their sum is constrained O_1+O_2+O_3+O_4 = N. This issue lead to further lack of comprehension regarding the validity of decomposing joint probability in line 122 to a product of variable-specific probabilities.

L101-104. I strongly advise the authors to be more explicit on what exactly they call “approximately distributed as Poisson distribution” and under what conditions their criteria of being approximately distributed is satisfied. Additionally, a general clarification of what motivates them to rely on such approximation would be useful.

L159-160. “The abundances of these two species are the same and given” is a very confusing sentence. Please clarify more rigorously.

Figure 1,2,3. The lack of sharpness these presence-absence maps does not improve them. Also, the font used in axis is clearly too small, so it shall either be remove altogether or increased appropriately if considered necessary.

Figure 2,3. Panels 2(a) and 3(a) are identical. Please consider better redesign of presented Figures and panels.

L278. It seems rather questionable statement that it is clear from Figure 4 that these two species exhibit co-occurrence.

L278. Common name “red oak” shall be replaced to Latin name (since Latin notation is used throughout this experiment).

L286. Probably 4(b) shall be replaced with 4(f) here.

Figures 4&5, 6&7. The tree location figures (4 and 6) are generally difficult to read due to unclear interpretation of different shades of grey. Furthermore, the analysis is conducted for the data truncated to the presence-absence patterns, so the need for figures 4&6 is further decreased.

---

## Round 0.2 · accepted · Accept

Dear Dr. Chang - you have addressed the two reviewers' comments and I now recommend publication.